# Classification of Elastic Wave for Non-Destructive Inspections Based on Self-Organizing Map

**Katsuya Nakamura \*** , **Yoshikazu Kobayashi, Kenichi Oda and Satoshi Shigemura**

Department of Civil Engineering, College of Science and Technology, Nihon University, Tokyo 101-8308, Japan
\* Correspondence: nakamura.katsuya@nihon-u.ac.jp

**Abstract:** An arrival time of an elastic wave is the important parameter to visualize the locations of the failures and/or elastic wave velocity distributions in the field of non-destructive testing (NDT). The arrival time detection is conducted generally using automatic picking algorithms in a measured time-history waveform. According to automatic picking algorithms, it is expected that the detected arrival time from low S/N signals has low accuracy if low S/N signals are measured in elastic wave measurements. Thus, in order to accurately detect the arrival time for NDT, the classification of measured elastic waves is required. However, the classification of elastic waves based on algorithms has not been extensively conducted. In this study, a classification method based on self-organizing maps (SOMs) is applied to classify the measured waves. SOMs visualize relation of measured data wherein the number of classes is unknown. Therefore, using SOM selects high and low S/N signals adequately from the measured waves. SOM is validated on model tests using the pencil lead breaks (PLBs), and it was confirmed that SOM successfully visualize the classes consisted of high S/N signal. Moreover, classified high S/N signals were applied to the source localization and it was noteworthy that localized sources were more accurate in comparison with using all of the measured waves.

**Keywords:** self-organizing map; arrival time; elastic wave; non-destructive testing; AE source localization

## 1. Introduction

In the field of non-destructive testing (NDT), parameters obtained from an elastic wave related with the failure process, for instance, the location of acoustic emission (AE) which the elastic wave generated by the occurrence of cracks sources, propagation velocities are useful for the index of sustainable existing structures. In the NDT using the elastic waves, AE source localization is attractive for visualizing micro cracks generated in infrastructures [1,2], and it has been used to evaluate the soundness of infrastructures. Moreover, elastic wave tomography [3] and AE tomography [4] have been applied to identify the elastic wave velocity distribution of materials [5,6] and it is expected that the location and size of inside failures in an infrastructure, is visualized. In the above NDT, elastic wave arrival time at the sensors used in the measurement is used to identify the failures. In addition, the arrival time is located in a boundary between noises and signals in a measured time-history waveform and can be detected based on a visual confirmation. However, in the arrival time detection, the use of a visual confirmation is challenging since a large number of AE are emitted during AE measurement [7–9]. Furthermore, the use of a large number of elastic waves is required to conduct the detailed identification of the velocity distribution because of the identification based on inverse problems. Therefore, the arrival time detection should be conducted using automatic picking algorithms.

In the algorithms of automatic arrival time detections, several approaches have been proposed to detect the accurate arrival time. The simple approach is a use of a threshold and if an amplitude exceeds the threshold, the exceeding time is defined as the arrival time. However, it is difficult that the original arrival time is detected by the threshold defined by

an amplitude level since the value of amplitude at arrival time is unknown. Thus, the use of characteristic functions instead of directly using amplitudes is proposed [10,11]. In other approaches, Akaike information criterion (AIC) has been applied to measured waveform to evaluate the boundary noises and signals [12,13]. It is noted that a determination of threshold considered the measurement conditions, is not required in AIC pickers to detect the arrival time. Although, theses conventional approaches can detect the accurate arrival approximated visual confirmation accuracies, it is challenging that the elastic wave arrival is detected from low S/N signals because decreasing the amplitude of the signal contributes to the low accuracy of the index computation to detect the arrival time. Since low S/N signals should not be used in NDT, the classification of elastic waves is required. However, eliminating low S/N signals is difficult. Arrival time detections conduct in waveforms and in order to measure waveforms, AE measurement systems generally use the threshold of the measurement trigger wherein waveforms are measured if the amplitude exceeds the threshold [14]. It should be noted that owing to the measurement trigger determination based on an empirical rule, there is a risk of including low S/N signals in the measurement data. Moreover, although high S/N signal is increased in measurement data if the measurement trigger is set higher values in comparison with the average of amplitudes of propagating waves, it is expected that the arrival time is not recorded in the waveforms. It should be noted that the measured waveform includes the pre-trigger time wherein is the duration from the start of the waveform to the time of the actuated measurement trigger, and the arrival time is located in the pre-trigger time. Hence, it is challenging to record the arrival time in the pre-trigger time since the start of measurement with the high measurement trigger is largely delayed from the start of signal. In the recent studies [9,15–17], elastic wave measurements have depended on the measurement trigger determined by empirical rules, and the classification of elastic waves based on algorithms has not been extensively conducted. Therefore, if the dependence of empirical rules in the classification is improved by algorithms, NDT conducts with accurate arrival times and it is expected that results of NDT are improved.

According to the above reasons, the measured waves include low S/N signals, and the classification of elastic waves is required for NDT using elastic waves. In this study, a classification method based on self-organizing map (SOM) [18] is applied to classify the measured waves to accurately detect the arrival time for NDT. SOM is a one of AI algorithms and SOM visualizes relations of measured data wherein the number of classes is unknown. In addition, SOM is categorized as an unsupervised learning method. If SOMs are applied to the classification of elastic waves, it is expected that SOMs improve the dependence of the empirical rule in the classification because the unsupervised learning does not require knowing characteristics of the data in the classification. Moreover, SOMs have been applied to the classification of AE signals generated in cross-ply composite specimens during tensile tests [19]. Therefore, it is expected that using SOM selects high and low S/N signals adequately from the measured waves. In order to validate the classification of elastic waves based on SOM, the model tests using the pencil lead breaks (PLB) are conducted, and classified high S/N signals generated by pencil lead breaks are applied to the source localization. Further, the specimen used in the model test, has a defect and the defect causes diffractions in wave propagations. The diffraction contributes to attenuate amplitudes of waves, and it is expected that high and low S/N signals are measured in the model test.

## 2. SOM Analysis

### 2.1. Architecture of SOM

In the algorithm of SOM, the class of the measured data is visualized in the output layer. The output layer as shown in Figure 1, is consisted of the neurons allocated with the weight vectors. The number of the neuron is generally determined by the heuristic rules.

In this paper, the heuristic equation used in an application of SOM [20], is applied to the determination of the number of the neurons *J* which is as follow:

$$J = 5\sqrt{N}, \tag{1}$$

where *N* is the number of input vectors obtained by measured data. At first phase of SOM, the weight vectors are set random values and the random values is iteratively updated by the input vectors for the visualization of classes. The updated weight vectors have similarity in the neighbor neurons. Therefore, it is noted that the class is visualized by the group of neurons obtained by the similarity of the weight vectors.

In order to update the weight vectors, a one of the weight vectors is selected by the Euclidean distance *d* obtained as

$$d_j = \|X_i - M_j\|_2, \tag{2}$$

where $X_i$ is the *i*th input vector, and the weight vector $M_j$ in the neuron *j*. Moreover, *x* is input vector component, *m* is weight vector component, $X_i$ and $M_j$ have the number of the vector components *n*. The Equation (2) is used to select the winning neuron which is minimum *d* in all of neurons. Furthermore, the neighbor of the winning neuron is updated by the Equation (3) obtained as

$$M'_j = M_j + h_c\left[X_i - M_j\right], \tag{3}$$

where $M'_j$ is the updated weight vector in the neuron *j*, and $h_c$ is the neighborhood function. It is noted that $M'_j$ is computed based on $h_c$ and the value of $h_c$ is obtained as

$$h_c = a(t)exp\left(-\frac{\|r_c - r_j\|_2^2}{2\sigma^2(t)}\right), \tag{4}$$

where $a(t)$ and $\sigma(t)$ are monotonically decreasing functions, $r_c$ and $r_j$ are the position vectors of the winning neuron *c* and the neuron *j*. In the Equation (4), $a(t)$ is defined as

$$a(t) = a_0 - \frac{t}{T}, \tag{5}$$

where $a_0$ is the initial value, *t* is the iteration count with $0 \leq t < T$, *T* is total iteration number. Moreover, $\sigma(t)$ is the same decreasing ration as $a(t)$ and $\sigma(t)$ is defined as

$$\sigma(t) = \sigma_0\left(1.0 - \frac{t}{T}\right), \tag{6}$$

where $\sigma_0$ is the initial value. In the Equations (5) and (6), while the value of $a_0$ is 1.0, the value of $\sigma_0$ is the initial radius of the update. The values of $a(t)$ and $\sigma(t)$, are, respectively, decreased with the iteration to prevent the diverges of the obtained classes.

According to the Equation (4), the maximum value of $h_c$ is obtained if $r_j$ is equal to $r_c$, and decreased with the distance between $r_j$ and $r_c$. It is worth noting that since the neurons are updated based on $h_c$, the group of neurons had the similarity of the weight vectors is appeared around *c*, there is the reason why SOM visualizes the class. In addition, the flow of SOM is iteratively conducting the Equations (2)–(4) with all of input vectors. Thus, the winning neuron is obtained by each input vector and the winning neurons form the several classes.

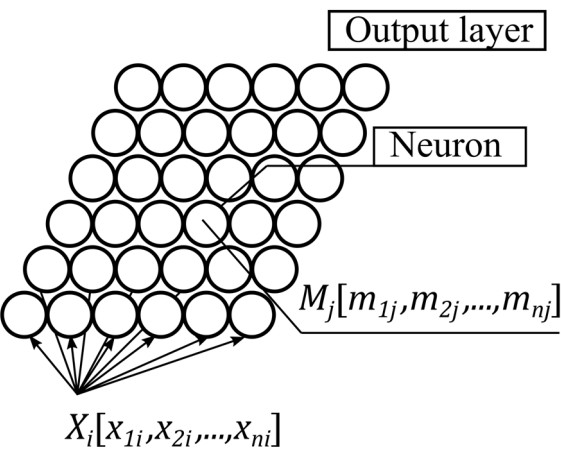

**Figure 1.** Architecture of an SOM.

### 2.2. Classification of Elastic Waves

The map which is the output layer visualizing the classes, can be used in the classification of the measured data if the closest Euclidean distance between the neuron and the measured data in the map is computed by the Equation (2) and the class of the winning neuron is known. Moreover, since the weight vectors in the class are interpolated by the winning neuron based on the Equation (4), the dispersion of measured data is considered in the map. Therefore, if an SOM forms the class in which is consisted of neurons updated by high S/N signals, it is expected that the obtained class selects the waves applied to NDT.

In this classification of the elastic waves, the input vector component is the root mean square voltage $V$ obtained from the equally divided waveform as shown in Figure 2. Although the number of the input vector component is related with the number of classes since the components imply the characteristics of measured data, the object of this classification is to select high and low S/N signal and it should be noted that the classification does not require obtaining the huge number of classes. Therefore, the input vector components should be decreased to prevent that the complex map is obtained. Furthermore, three components have been used in the classification of using RGB value [21] and it implies that the use of 3-dimensions vectors is sufficient in the simple classification. Hence, it is expected that the use of $V_1$, $V_2$, and $V_3$ for the input vector components contribute to the simple classification.

The boundary of the classes is not cleared in the obtained map and the boundaries have been indicated by the computing similarity of the weight vectors [19,22]. In this study, since the boundary are easily determined, the three types of the weight vectors are determined using the ration of the root mean square voltage between the pre-trigger time and the signal duration in all of the neurons, and based on the Equation (2), the 3 types of classes are obtained by the winning neurons. In the measured waveform shown in Figure 2, 0 μs implies the time of a measurement wherein the amplitude exceeds the measurement trigger. Moreover, the negative time is the pre-trigger time and the positive time is the signal duration. Further, it should be noted that the ration of the root mean square voltage implies the level of S/N because the high S/N signal tend not to be measured in the pre-trigger time. Therefore, it is possible that the neurons had the highest ration of the root mean square voltage in the map belongs to the high S/N class. Moreover, the low S/N signal is assumed to be classified between attenuated elastic waves and noises: for instance, reflection waves, electrical noise, other noises generated in outside of the materials. The noises generally have lower amplitudes in comparison with other measured waves, and it is expected that the neurons had the lowest voltage ration belongs to the noise class in which the noises are classified. Therefore, the weight vectors used for the criterions of classes are selected from the neurons had the highest and the lowest voltage. Furthermore, the low S/N class in which the attenuated elastic wave is classified is determined by the average vectors computed by two of other weight vectors used for the criterion.

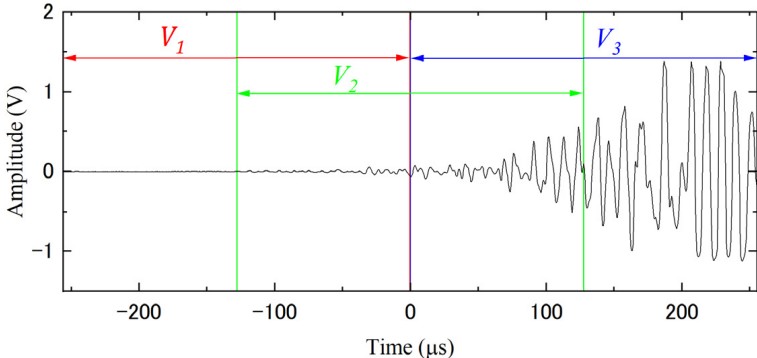

**Figure 2.** Illustration of the root mean square voltages obtained from the equally divided waveform.

## 3. Experimental Set Up

### 3.1. Artificial AE Measurement Conditions

In order to validate the performance of the wave classification based on SOM, high and low S/N signals should be measured in the model test. According to Huygens-Fresnel principle, amplitude of the diffraction waves is lower in comparison with the wave propagated in the straight line. Therefore, it is expected that high and low S/N signals could be measured if the diffraction waves are generated in the model test. In the model test, an aluminum plate with a thickness of 5.0 mm, is used as the specimen. The aluminum plate shown in Figure 3, is square-shaped with a side of 1000 mm, and the defect is 40 mm and 500 mm in height and width, respectively. Furthermore, 12 of the sensors are installed 10 mm inside the frame. It is noteworthy that the diffraction waves propagate around the defect, and it is expected that the diffraction wave is measured in Ch3 and Ch4, if artificial AE is generated in the upper side of the specimen. In addition, artificial AE is generated by PLB, which is widely used in studies of AE source localizations [23–25]. In PLB test, pencil lead is a diameter of 0.5 mm and hardness of 2H. The measurement conditions listed in Table 1 refer the previous study conducting PLB [26]. In the measurement, the AE measurement system produced by PAC [14] is used. The operating frequency of R6a was 35–100 kHz, and the sensors have been applied to metal materials to measure the waves generated by PLB [26,27]. Further, the measured artificial AE is amplified by 40 dB via the 2/4/6 preamp, and the sampling frequency on the measurement boards of the Express-8 is 2 MHz. In addition, the wavelength is measured as 511.5 µs if the amplitude exceeded 55 dB of the threshold, and the pre-trigger time is set to 256 µs.

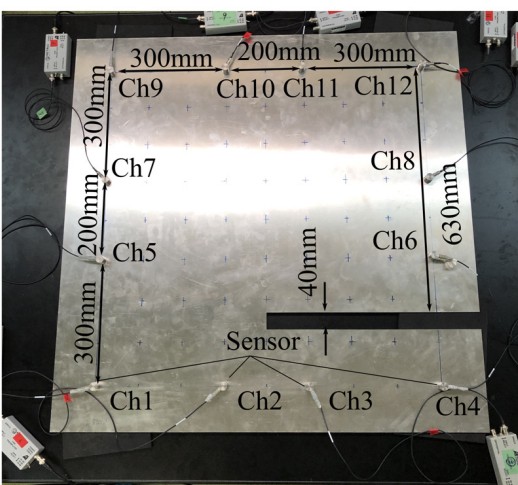

**Figure 3.** Aluminum plate used in the model test and sensor locations.

**Table 1.** The measurement conditions of the model test.

| | |
|---|---|
| Operating frequency (kHz) | 35–100 |
| Preamplifier gain (dB) | 40 |
| Sampling frequency (MHz) | 2 |
| Threshold of measurement trigger (dB) | 55 |
| Pre-trigger time (μs) | 256 |
| Wavelength (μs) | 511.5 |

*3.2. Arrival Time Detection*

A one of AIC pickers using a single parameter is AR-AIC [12] and the arrival time detection only requires the number of the AR model order. Hence, it is expected that a use of the single parameter contributes to improving the differences of arrival time accuracy caused of users. Moreover, AR-AIC have been applied to the source localization in geomaterials had higher attenuation in comparison with other construction materials [28]. Therefore, AR-AIC is applied to the classified waves and the detected arrival times are used in the AE source localization in this study.

The example of arrival time detection using AR-AIC, is shown in Figure 4. In AR-AIC, the minimum AIC value is the index of the arrival time. The arrival time detections based on AIC assume that measured wave can be approximated continuous functions and AIC pickers detect a large amplitude difference. Thus, if an amplitude of first arrival is particularly lower in comparison with other amplitudes, it is possible that the arrival time accuracy is decreased [13]. In use of AR-AIC, owing to the above problem, the elastic waves should be classified to detect accurate arrival times.

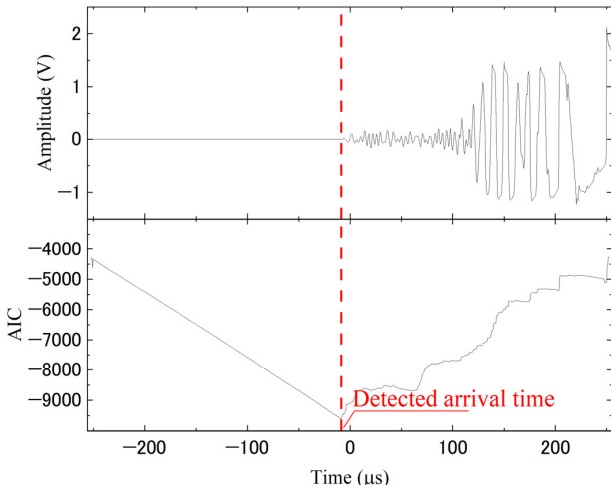

**Figure 4.** Illustration of the arrival time detection method using AR-AIC.

*3.3. Validation of the Classification Using SOM*

In this validation of the classification, if the measured waves at Ch4 are classified from all of waves, SOM is expected to classify the measured waves for NDT. PLB is conducted around Ch9 as shown in Figure 5. Moreover, PLB is repeatedly conducted 10 times, and AR-AIC is applied to all of the measured wave to detect arrival times at each sensor. If the artificial AE signal generated by PLB propagates to all of the sensors, the diffraction waves are measured at Ch3, Ch4 and the straight waves arrive at the other sensors. In the model test using an aluminum plate with a thickness of 5.0 mm, the artificial AE signal propagates as Lamb waves. Phase velocity of Lam wave has the frequency dispersion of velocity. According to the phase velocity dispersion curves for an aluminum plate with a thickness of 5.0 mm shown in Figure 6, the phase velocities of the S0 modes are approximately constant at the highest operating frequency shown in Table 1. Therefore, if travel times are computed by subtracting the arrival time at Ch9 from the other arrival times, the velocities computed by the ration of the theoretical ray-paths and the computed

travel times approximate the phase velocity of S0 mode lamb wave in which is 5400 m/s. The average of the computed velocities is shown in Figure 7, and it is confirmed that the average of the velocities using the arrival times at Ch4 is different from the phase velocity of S0 mode lamb wave. Therefore, it is possible that the arrival times at Ch4 include arrival detection errors and the measured waves at Ch4 should not be used in NDT.

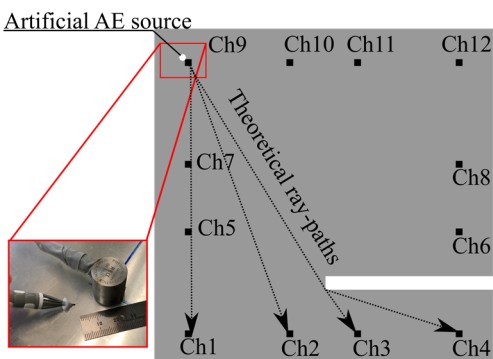

**Figure 5.** The method of generating artificial AE and illustration of theoretical ray-paths from the source to each the sensor.

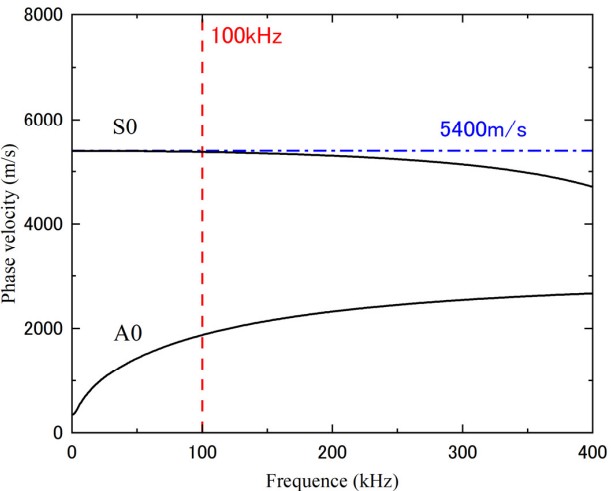

**Figure 6.** The phase velocity dispersion curves for an aluminum plate with a thickness of 5.0 mm.

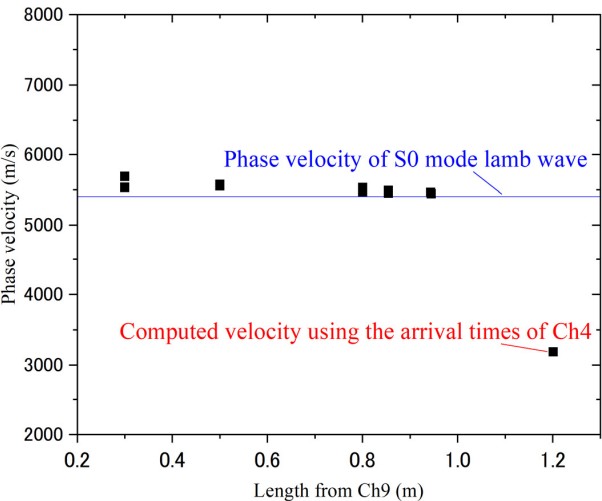

**Figure 7.** The average of the velocities computed by travel times from Ch9 to other sensors.

Although a sufficiently large number of total iterations are required in an SOM analysis [18], generally, the number is empirically determined. In this study, the number of total iterations is 10,000, and the parameter is validated by the results of the classification. Moreover, in order to initially update sufficient neurons, the initial radius uses one-third of one side of the output layer which is a square. According to Equation (4), the update of neurons is obeyed by Gaussian distributions. Hence, the used initial radius is expected to update more than half of neurons if the winning neuron is located in edge of the output layer.

### 3.4. AE Source Localization with the Classfied Artifical AE Signals

To validate the performance of the elastic wave classification for NDT, the classified waves by SOM are applied to the AE source localization based on ray-tracing [29]. Further, SOMs use limited waves at the sensor which measures the largest number of waves in the model test to validate the practical classification. If an SOM is conducted with limited data to visualize classes and it is expected that the computation cost of SOM is conserved.

The AE source is localized on the basis of the possible pulse-originated time computed by using the first travel times computed by ray-tracing. Further, in the ray-tracing algorithm, the elastic wave velocity distributions of the models are approximated by the mesh in which each cell has a constant velocity, and first travel times considered diffractions of waves are computed by the approximated elastic wave velocity distributions. The possible pulse-originated time $P_{ij}$, is computed by subtracting the first travel time from the arrival time as follows:

$$P_{ij} = A_i - Tr_{ij}, \tag{7}$$

where $A_i$ is the arrival time of the AE sensor $i$, $Tr_{ij}$ on the ray-path from the sensor $i$ to the candidate $j$. The number of estimated originated times is the same as the number of used sensors at all candidates because the computation of the originated time is conducted on all of the combination of a sensor and a candidate. A candidate in which variance of the originated time $\sigma_j^2$ is minimum is selected as the AE source. The localized source is illustrated in Figure 8. It is noted that $\sigma_j^2$ can be computed by the limited sensor. Therefore, in the AE source localization based on ray-tracing, the sensors which the accurate arrival times are detected can be selected based on the result of classifications. The time difference of arrival (TDOA) method [30] is a popular AE source localization method in the field of NDT. However, the TDOA method requires computing matrix equations and the determinant is possibly approximated 0 with the particular location of the used sensors. Furthermore, the TDOA method generally assumes homogeneous elastic wave velocity distributions, it is challenging that the TDOA method is applied to this heterogeneous model. On the other hand, the ray-tracing can consider diffraction and refraction waves occurred in heterogeneous velocity distributions and it is expected that the AE source localization based on ray-tracing can localize the source generated in heterogeneous model.

In this validation, artificial AE sources generated by PLB, are localized. PLB is repeatedly conducted with the interval of PLB points 100 mm as shown in Figure 9. Thus, it is expected that total 49 events are localized in the model test. AE source localization based on ray-tracing is conducted in the approximated velocity distributions of the model. The approximated velocity distributions for the specimen are show in Figure 10a. In the model show in Figure 10a, the original point of the orthogonal coordinate system is defined as Ch1 and the defect in the specimen is shown by low velocity distributions and the source candidates are located with the interval of candidates 20 mm. The propagation velocity in the soundness area is 5400 m/s, wherein the phase velocity of S0 mode lamb wave is applied. Moreover, the velocity distributions in the defect are assumed to be velocities of air 300 m/s. If the classification of artificial AE signal is practically conducted, the accuracy of the localized source is improved in comparison with the sources localized by all of waves. Furthermore, if SOM has the performance to classify diffraction waves, it is expected that the AE source localization based on ray-tracing can be conducted in homogenous distributions shown in Figure 10b.

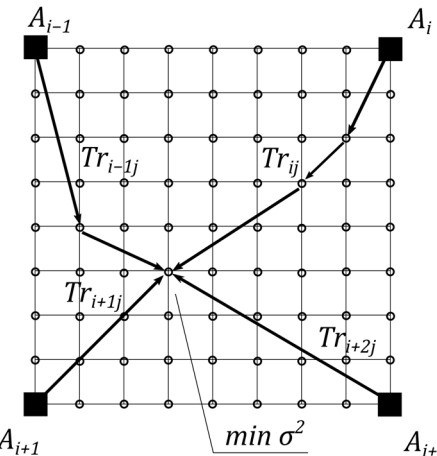

**Figure 8.** The conceptual diagram of the localized AE source.

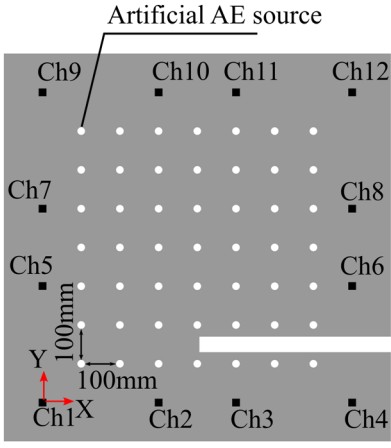

**Figure 9.** Aluminum plate with PLB points.

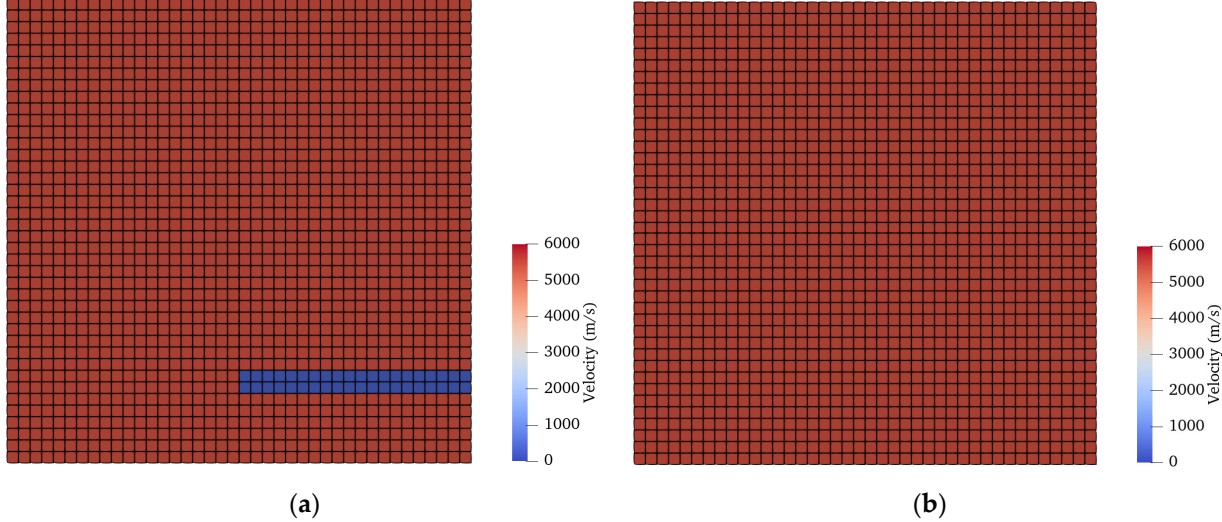

**Figure 10.** The velocity distributions used in AE source localization based on ray-tracing: (**a**) the approximated velocity distributions for the specimen; (**b**) the homogenous velocity distributions.

## 4. Results

### 4.1. Results of the Classification Using SOM

The result of the SOM, which is the map using all of measured waves, is shown in Figure 11. It should be noted that the map is same the structure as the output layer shown in Figure 1. In Figure 11a, each cell is the neuron and the center of cells is allocated with the weight vectors. Moreover, the number of neurons is determined based on Equation (1) with 168 waves, and the result of Equation (1) is approximated 64 for the map formed rectangular structure. Although 120 artificial signals are theoretically measured in the model test because the PLB is conducted 10 times with 12 sensors, total measured waves are 168. Thus, the noises are measured in the model test and the map is required to classify the noises in the noise class. Colors shown in the map imply classes of waves, and the red area is the high S/N class. In addition, the gray and the blue area imply the low S/N class and the noise class. Further, the gray scale shown in Figure 11b implies how many waves belong to each cell. If classified waves in the red are only consisted of high S/N signals in which accurate arrival times are detected, the classification based on SOM has the potential to select high S/N signals for NDT. In order to detail the performance of the classification, the class of measured waves in each sensor is shown in Figure 12. In each sensor, the number of the measured waves belonged to the high and low S/N class is totally 10 times and the number is the same as the number of PLB test times. Therefore, it is confirmed that the map has the performance to classify between the noise and the artificial AE signal. Although detecting accurate arrival time from the wave measured at Ch4 is challenging, and waves measured at Ch4 should be classified in the low S/N class or noise classes, artificial AE signals measured at Ch4 are classified in the low S/N class in Figure 12. Furthermore, according to Figure 6, the classified waves in the high S/N class can be detected accurate arrival times. Hence, it implies that the classification based on SOM, is possible to select high S/N waves in which the accurate arrival time is detected for NDT. In addition, it is confirmed that used parameters in SOM, are sufficient values to classify the measured waves using 3-dimension input vectors.

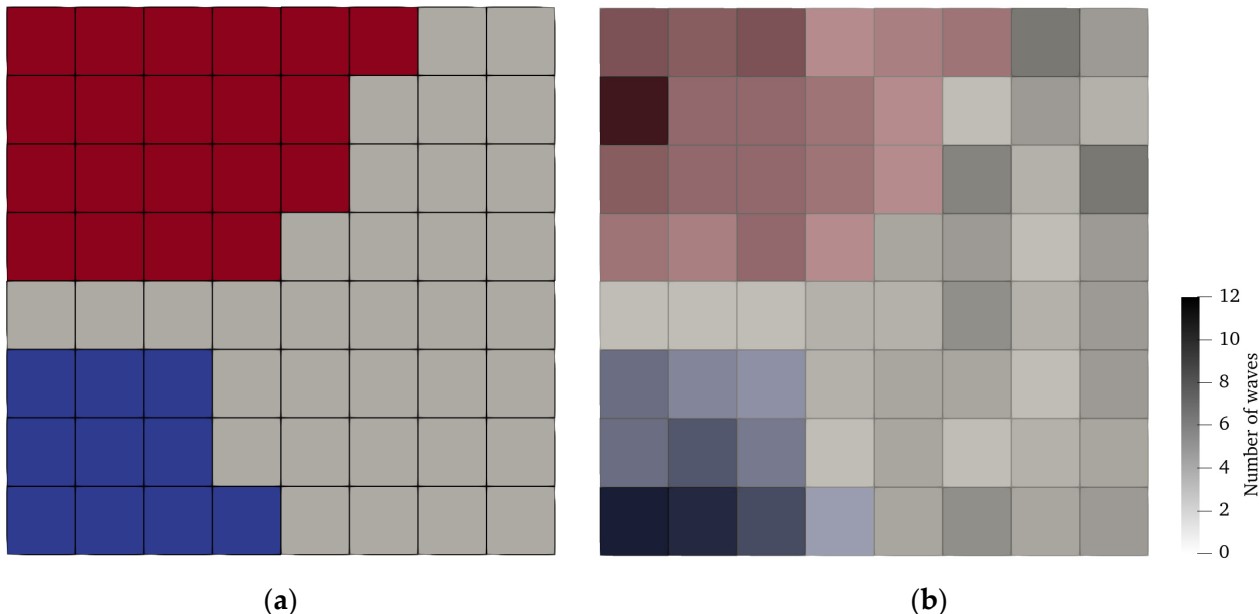

(a)                  (b)

**Figure 11.** Results of SOM: (**a**) visualized classes on the map; (**b**) the number of measured waves belonged to each neuron.

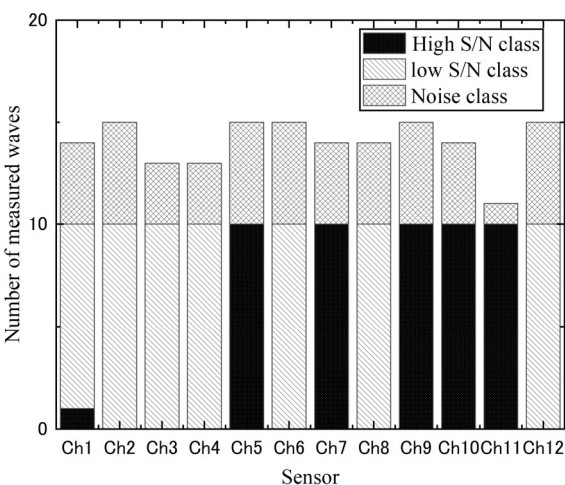

**Figure 12.** The number of classified waves at each sensor.

### 4.2. Results of AE Source Localization with Classified Artificial AE Signals

In measured waves at Ch6, it is expected that three of the classes are included because the number of measured waves at Ch6 is largest in the model test. Thus, measured waves at Ch6, are applied to SOM and the result is shown in Figure 13a. It should be noted that 56 waves are measured in Ch6 and the map is consisted of 36 neurons which is the approximated value of Equation (1). According to Figure 13a, it is confirmed that the use of all of waves is not required to update the map since 3 of classes are visualized on the map updated by limited waves. In Figure 13b, all of measured waves are classified in 3 of classes, and it implies that the map can be used to classify all of measured waves in each class. In the number of classified waves shown in Figure 14, the large number of measured waves at Ch3 and Ch4 are classified in the low S/N and the noise class. According to Figure 9, diffraction waves propagate from PLB points located in upper side of the specimen to Ch3 and Ch4. According to Figure 12, diffraction waves arrived at Ch4 tend to be classified in the low S/N signals. Thus, it implies that low S/N signals can be classified on the map, and it is expected that the map contributes to improve the results of AE source localization with classified waves. The results of AE source localization based on ray-tracing are show in Figure 15. While the sources localized with all of measured waves are shown in Figure 15a, the results of AE source localization with classified artificial AE signals are shown in Figure 15b. Moreover, it should be noted that white dots imply the location where PLB test is conducted and if the localized source which is the black dot is located on the white dot, the accurate source is localized. In Figure 15b, the number of the localized source in original PLB points are increased in comparison with the results shown in Figure 15a. In addition, the map is expected to classify the diffraction waves. In order to validate the performance of the classification in the homogeneous model, the source localization with classified artificial AE signals is conducted in homogeneous velocity distributions shown in Figure 16. According to Figure 16, it is confirmed that the number of localized sources on the original PLB points is 38 sources the same as the results in the heterogeneous velocity distributions. The performance of the classification in the AE source localization are shown in Table 2. Although the number of total sources with the non-classification is 54 sources, the number of localized sources with SOM is 49 sources the same as PLB tests. In the non-classification, it is expected that five sources are generated by noises because the number of PLB tests is 49 time in the specimen. Hence, it implies that SOM has the performance to eliminate non-target sources generated by noises. Further, it is confirmed that the maximum and the average of errors in the use of SOM are smaller in comparison with the non-classification. In the non-classification case, it should be noted that the non-target sources are eliminated in the maximum and the average of errors. Furthermore, the maximum and the average of errors in the source localization with classified artificial AE signals are approximated in heterogeneous and homogeneous velocity distributions. Therefore, if SOM is applied

to the AE source localization based on ray-tracing, it implies that the consideration of heterogeneity in materials is not required to localize sources.

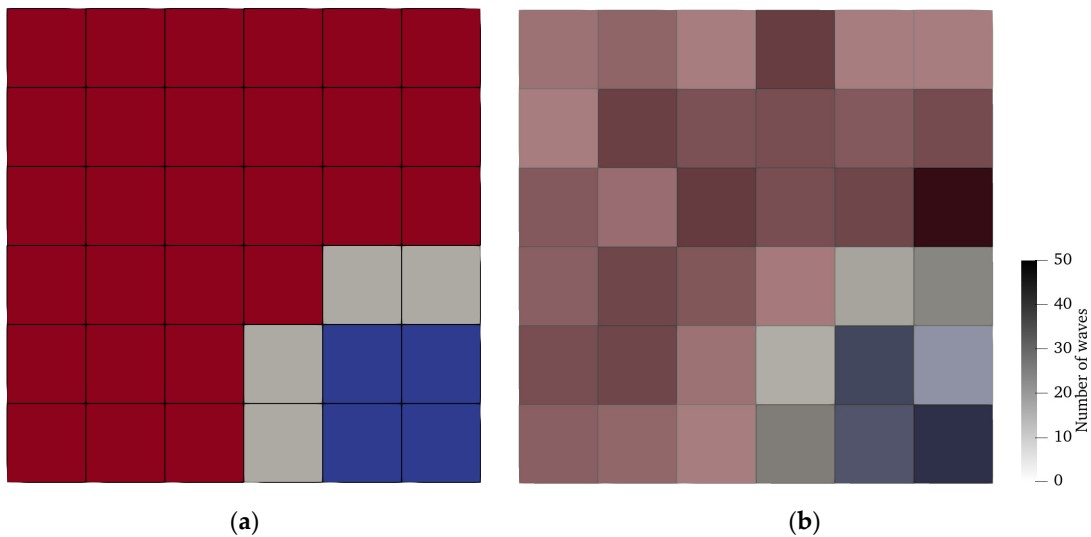

(**a**)  (**b**)

**Figure 13.** Results of SOM computed by measured waves at Ch6: (**a**) visualized classes on the map; (**b**) the number of measured waves belonged to each neuron.

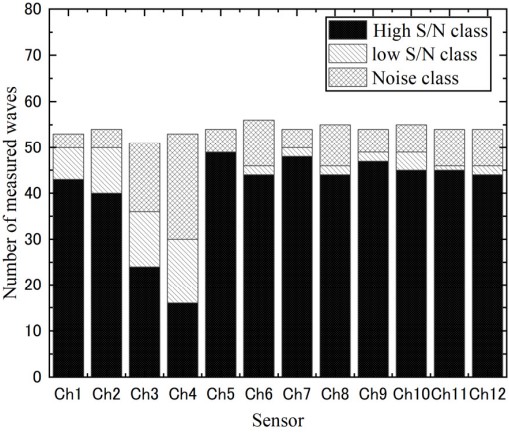

**Figure 14.** The number of classified waves based on the map updated by waves measured at Ch6.

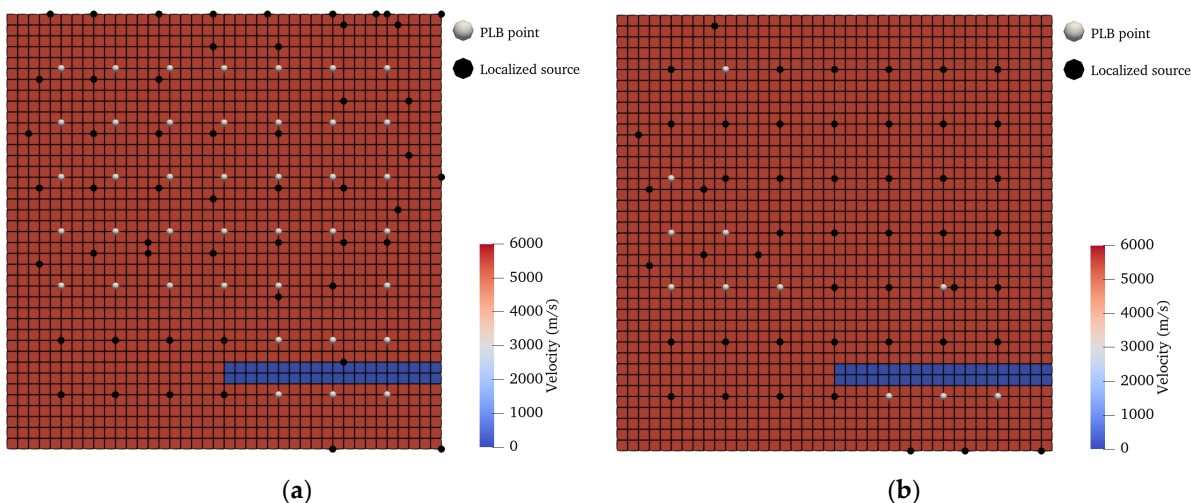

(**a**)  (**b**)

**Figure 15.** Results of AE source localization based on ray-tracing: (**a**) the source localization in non-classification; (**b**) sources localized by classified artificial AE signals.

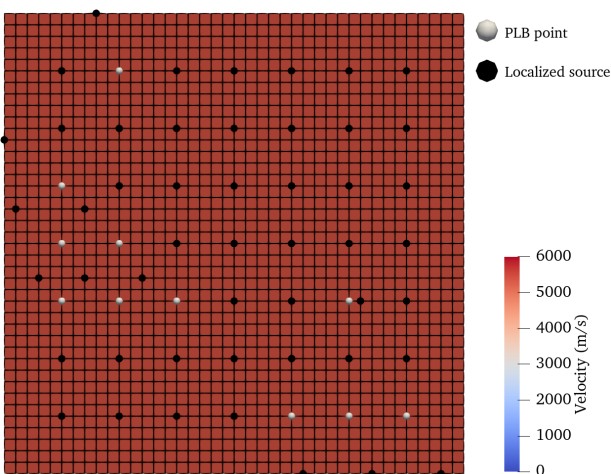

**Figure 16.** Results of the AE source localization in homogeneous velocity distributions.

According to the algorithm of AE source localization based on ray-tracing [29], the factors of errors in the AE source localization are the locations of candidates of sources, considered diffractions, and arrival time detection errors. In this model test, candidates of sources are same the locations as PLB points and the locations of the candidates did not contribute to localization errors. In the heterogenous velocity distributions, the AE source localization considers the diffraction and it is expected that the factor of velocity distributions is limitingly affected in the maximum error listed in Table 3. Hence, arrival time detection errors are expected to contribute to the maximum localization error in the heterogenous velocity distributions. Although the diffraction is not considered to localize sources in the homogenous model, arrival times of diffraction waves are used in the AE source localization conducted on the homogenous model. Thus, it implies that the diffractions contribute to localization errors in the homogenous model. However, according to the Table 2, the average of localization error in the homogenous model are approximated to the heterogenous model and it is expected that considered diffractions are not the main factor of the maximum error. Further, in the localized sources with maximum errors, the wave is propagated with the larger angle of diffraction to Ch7 in the heterogenous model, and Ch4 measures the diffraction wave used in the homogeneity model. Moreover, the arrival time detection in diffraction waves is shown in Figure 17. The waveform shown in Figure 17a is measured at Ch7, and Figure 17b shows the measured wave at Ch4. In Figure 17, the arrival times are unclear, and it is difficult that the arrival times are detected by visual confirmations. Furthermore, applied AR-AIC detects times at middle of waveforms in Figure 17, it is confirmed that detected arrival times at Ch7 and Ch4 include detection errors. In addition, detection errors are confirmed in other waveforms for the AE source localization and it is confirmed that detection errors contribute to the source localization error in the model test. Therefore, it implies that the high S/N class includes low S/N signals in which arrival times are detected with the low accuracy.

**Table 2.** The performance of the classification in the AE source localization.

| Classifications | Velocity Distributions | Events Used | Number of PLB Tests | Maximum Errors (mm) | Average of Errors (mm) | Number of Accurate Sources |
|---|---|---|---|---|---|---|
| Non-classification | Heterogeneous | 54 | 49 | 220 | 90 | 9 |
| SOM | Heterogeneous | 49 | 49 | 130 | 19 | 38 |
| SOM | Homogeneous | 49 | 49 | 130 | 20 | 38 |

**Table 3.** Maximum errors and number of sensors used in the AE source localization.

| Velocity Distribution | Actual PLB Points (mm) | | Localized Sources (mm) | | Maximum Errors (mm) | Number of Sensors | The Diffraction Wave Arrival |
| --- | --- | --- | --- | --- | --- | --- | --- |
| | X | Y | X | Y | | | |
| Heterogeneous | 700 | 100 | 780 | 0 | 130 | 6 | Ch5, Ch7 |
| Homogeneous | 100 | 500 | 0 | 580 | 130 | 12 | Ch4 |

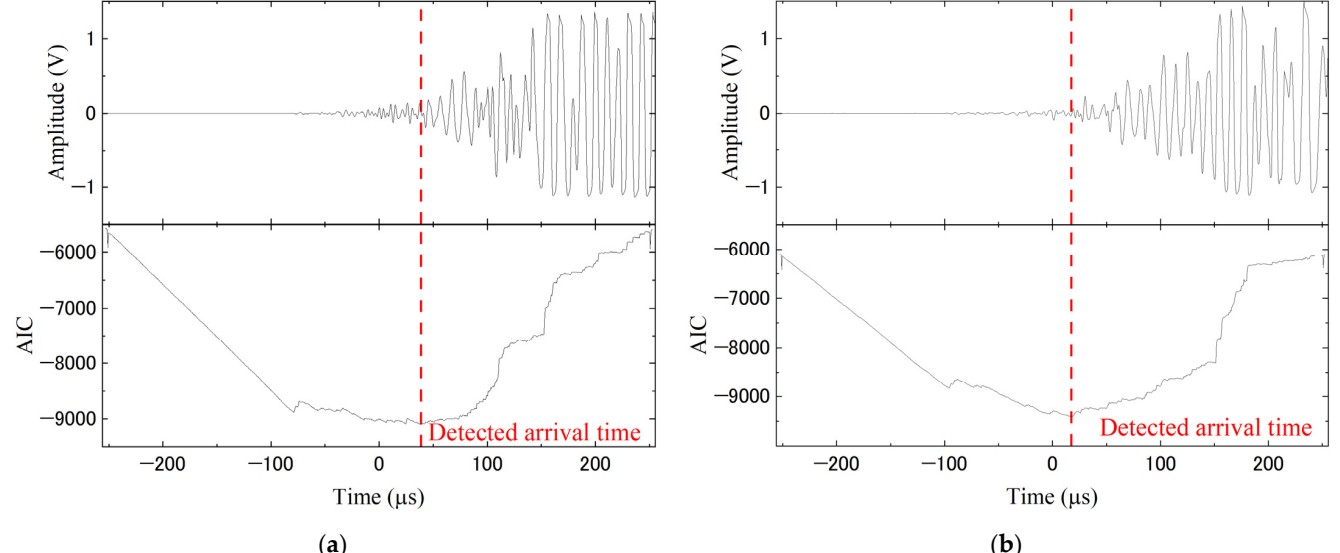

(**a**)                                      (**b**)

**Figure 17.** Arrival time detection in diffraction waves: (**a**) Waveform measured at Ch7; (**b**) Waveform measured at Ch4.

## 5. Discussion

Since a threshold of a measurement trigger in which is determined by empirical approaches is generally used to detect the waveforms, it is challenging to eliminate noises from measurement data. Particularly, AE measurement system requires to use resonant sensors [14] and the determination of the threshold should be considered differences of amplitudes depended on the resonate frequency and the size of an amplitude in the propagated waves. Thus, the determination of the practical measurement trigger requires to consider the combination between materials and the measurement system. In the elastic wave measurement, the noise is assumed to be reflected elastic waves, noises transmitted from outside of a specimen and electrical noises. According to the acoustic impedance, amplitudes of reflected and transmitted waves are expected lower in comparison with elastic waves since incident waves are separated between reflected and transmitted waves. Further, the duration of electrical noises is generally shorter than signals, and it is expected that the root mean square voltage obtained from electrical noises smaller in comparison with the voltage obtained from elastic waves. Therefore, SOM has the potential to classify measured waves based on differences of the root mean square voltage. According to Figures 12 and 14, it is confirmed that SOM performs to classify the artificial AE signals and the noises. Hence if the measurement data includes several noises because the measurement trigger is not appropriate for measurement conditions, SOM performs to eliminate noises and it implies that the dependency of the measurement trigger in the accuracy of measurements is improved. Moreover, the number of the input vector components imply the characteristic of data, and huge number of the components causes to form several scattered classes in the map. In this classification, the waveforms are not directly applied to input vectors and 3-dimensions input vectors in which components are consisted of the root mean square voltage obtained from the equally divided waveform are used. As results of using 3-dimensions input vectors, 3 of classes are formed in the map shown in

Figures 11 and 13, it is expected that 3 components are sufficient to classify elastic waves because the classes are not scattered.

In the results of the classifications shown in Figure 12, it implies that SOM has potential to classify waveforms based on the root mean square voltage. Furthermore, the value of the root means square voltage computed by the waveform is related with the levels of the amplitudes. According to Huygens-Fresnel principle, the amplitudes of diffraction waves are more attenuated in comparison with straight propagations wave. Thus, SOM is expected to separate elastic waves between the straight and the diffraction. According to Table 2, the accuracy of the source localization with the classified waves in homogeneous velocity distributions are approximated the results of the source localization considering heterogeneous velocity distributions. Therefore, SOM performs to classify the straight and the diffraction waves, and it is expected that the consideration of the diffractions caused by the heterogeneity of the material is not required in the source localization in heterogeneous velocity distributions. If an AE source localization is applied to an evaluation for soundness of infrastructures, sources are possibly generated in failure areas in which is lower elastic wave velocities than soundness areas. Therefore, it is expected that the accuracy of the evaluation of the soundness based on the AE source localization is improved if the AE source localization is conducted with the classified elastic waves. Further, although the huge number of the elastic waves should be measured in order to identify the process or locations of failures by NDT, the computation cost in SOM is increased with the number of waves. However, according to Figure 13, the classification based on SOM performs to visualize 3 of classes with a limited number of waves. It is expected that the dispersion of the measured waves is considered by the neighbor function defined as the Equation (4) and the missing data is interpolated in the map. In addition, the AE source localization is improved by the accurate arrival times detected from the waveforms classified by the formed high S/N class. Hence, it is confirmed that a result of SOM is possible to be applied to other measurement data if the data are measured in the same conditions as forming the map. Therefore, it is expected that the computation cost ca be more conserved in comparison with conducting SOM in each measurement if the formed map is shared in each measurement. Moreover, owing to the high S/N class shown in Figure 13 included diffraction waves in which detected arrival times are low accuracies, 11 sources are localized with localization errors in Figures 15b and 16. Since measured waves are not gathered in the center of the high S/N class shown in Figure 13b, it implies that the high S/N class can be divided. Thus, it is expected that if the high S/N class is divided in 2 classes and high S/N signals are selected based on the divided high S/N class, the accuracy of the source localization is improved in comparison with the results of the AE source localization shown in Table 2.

The classification based on SOM is expected to improve the identifications of elastic wave velocity distributions because the accurate arrival times are required in the identifications. The identifications of the velocity distributions require to obtain difference travel times computed subtracting the computed travel time form the measured travel time measure of diffraction and/or refraction waves [3–6]. In Figure 13a, the high S/N class includes diffraction waves, and it is expected that the classified diffraction waves are larger S/N in comparison with other diffraction waves. Thus, the classified diffraction wave has potential to be detected accurate arrival times. However, applied AR-AIC detects arrival times of the diffraction waves including detection errors. Therefore, it is expected that other arrival time detection method in which have detected arrival times from diffraction waves [26], is required to be applied to the classified waves in order to identify an accurate velocity distribution.

## 6. Conclusions

In order to conduct NDT in which are AE source localizations, elastic wave tomography, and AE tomography, accurate arrival times of elastic waves used for input data are required. According to the algorithms of the arrival time detection, the accuracy of the

detected arrival time depends on levels of S/N in measured wave forms. Hence, high S/N signal is required to be applied to the arrival time detections. Although a measured wave form is generally selected by thresholds of the measurement trigger, the determination of practical measurement triggers requires an empirical rule. An SOM is categorized as an unsupervised learning method. Thus, if an SOM is applied to the classification of elastic waves, it is expected that the SOM improves the dependence of the empirical rule in the classification because the unsupervised learning does not require knowing characteristics of the data in the classification. In this study, the classification of elastic waves based on SOMs for NDT is validated by the model tests. In the model tests, PLB tests are conducted in the specimen in which the aluminum plate with a thickness of 5.0 mm. Here, the conclusions of this study are listed as follows.

- According to Figure 12, in each sensor, the number of the measured waves belonged to the high and low S/N class is totally 10 times and the number is the same as the number of PLB test times. Therefore, it was confirmed that the classification based on SOM can performs to classify artificial AE signals and noises from measurement data.
- In this classification, waveforms were not directly applied to input vectors and 3-dimensions input vectors in which components were consisted of the root mean square voltage obtained from the equally divided waveform were used. According to the results of SOM, it was confirmed that artificial AE signals were classified by 3-dimentions input vectors computed based on the root mean square voltage.
- The AE source localization based on ray-tracing was conducted with classified waves. As consequence, the localized sources were more accurate in comparison with the use of all waves. Therefore, if the measurement data include several noises because the measurement trigger is not appropriate for measurement conditions, the SOM performs to eliminate noises and it implies that the dependency of the measurement trigger in the accuracy of measurements is improved.
- The accuracy of the source localization with classified waves in homogeneous velocity distributions were approximated the results of the source localization considered heterogeneous velocity distributions. Therefore, it is expected that the source localization in heterogeneous velocity distributions does not require considering the diffractions caused by the heterogeneity of the material if classified waves are used in the source localization.
- According to Figure 13, the classification based on SOM performed to visualize 3 of classes with a limited number of waves. In addition, the AE source localization was improved by the accurate arrival times detected from the waveforms classified by the formed high S/N class. Hence, it was confirmed that a result of SOM is possible to be applied to other measurement data if the data are measured in the same conditions as forming the map. Therefore, it is expected that the computation cost can be more conserved in comparison with conducting SOM in each measurement if the formed map is shared in each measurement.
- In Figure 13, the high S/N class included diffraction waves, and it is expected that the classified diffraction waves are larger S/N in comparison with other diffraction waves. Thus, the classified diffraction wave has potential to be detected accurate arrival times. However, applied AR-AIC detected arrival times of the diffraction waves including detection errors. Therefore, it is expected that other arrival time detection method in which has the potential to detect arrival times from diffraction waves, is required to be applied to the classified waves in order to identify an accurate velocity distribution by the tomography methods.

**Author Contributions:** Conceptualization, K.N.; methodology, K.N.; software, K.N.; validation, K.N.; formal analysis, K.N.; investigation, K.N., Y.K., K.O. and S.S.; resources, Y.K. and K.O.; data curation, K.N.; writing—original draft preparation, K.N.; writing—review and editing, Y.K., K.O. and S.S.; visualization, K.N. All authors have read and agreed to the published version of the manuscript.

**Funding:** This research received no external funding.

**Institutional Review Board Statement:** Not applicable.

**Informed Consent Statement:** Not applicable.

**Data Availability Statement:** Not applicable.

**Conflicts of Interest:** The authors declare no conflict of interest.

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
