# Peer review of "Classification of Elastic Wave for Non-Destructive Inspections Based on Self-Organizing Map"

_sustainability, doi:10.3390/su15064846_

Round 1

Reviewer 1 Report

In the field of non-destructive testing (NDT), arrival time detection is usually performed using an automatic picking algorithm in the measured time-history waveform. To solve the problem of low accuracy of detected arrival time in low signal-to-noise ratio signals. Classification of the measured elastic waveforms is required. The manuscript uses a classification method based on self-organizing mapping (SOM) to classify the measured waves, and the effectiveness of the method is verified by examining the classification results. The SOM model was validated using a pencil core break, and it was verified that the SOM can successfully visualize classes composed of high SNR signals. However, the manuscript still has the following problems that need improvement.

Comment 1: About the " Abstract": The abstract section is not clear enough to summarize the work and contributions of this manuscript.

Comment 2: About the " Introduction": The introduction section does not provide sufficient research on the existing research findings. The key scientific questions that need to be addressed are not reflected in the introduction.

Comment 3:About the " Conclusions": The conclusion is too simple and lacks a summary of this research. Strengthen the interpretation of its importance.

Comment 4: Figure should be unified standard and made more beautiful.

Comment 5: Checking the format of charts, references, etc. in the manuscript.

Comment 5: The discussion section in the manuscript is not in-depth enough.

Recommendation:this manuscript may be accepted after major revision.

Reviewer 2 Report

Clarifications on the points raised below ought to be provided before a definitive proposal regarding publication of the paper is made:

1.        Where do your study findings have a practical application? This particular question is related with the problem statement of your study.

2.        Very limited recent research is included in the manuscript and the same has been reflected in the reference section.

3.       The study does not contain more number recent literatures.

4.         Research significance / research gap is missing in introduction section.

5.         The manuscript does not go into considerable detail about how to apply and discuss the findings. The conclusions are similarly lacking in information.

6.         Figure 7 is not available. After figure 6, it is mentioned as figure 8.

7.         Which are all the factors affecting the maximum errors (mm) and number of sensors in Table 3?

8.         Discuss in detail about the Figure 11 and 13 in text for better understanding to the readers.

9.         Throughout the technical paper, the results are compared and not discussed in a detailed manner with mechanism for changes in the test results for Velocity distribution.

Reviewer 3 Report

The authors of the submitted manuscript presented an approach of elastic waves classification based on self-organizing maps (SOM) to eliminate noises from measured AE signals. The study is interesting, and solves an important problem within the field of NDT data conditioning. The presented results are original and the manuscript worth further consideration, however, numerous improvements are necessary according to the detailed comments below.

1.     The performed literature survey requires extension: SOM is the tool used by the authors for classification, however, the authors did not considered it in the literature survey. It is thus recommended to perform an overview on other classification approaches and SOM, provide applicability of SOM in similar problems (damage/fault detection/classification), and link this survey with the formulation of a motivation.

2.     The motivation of the study should be emphasized, in particular, the selection of SOM for solving the problem should be justified.

3.     The considered damage in the plate is very big. The question is how the developed approach is dealing with lower damage, e.g. cracks, and what is its lower limit of detectability. This will define the practical applicability of the approach.

4.     Please provide detailed information on experimental setup, i.e. type of sensors, other equipment used during the experiment, including its models and manufacturers. This will make the performed experiments repeatable for the readers.

5.     In section 3.3, the used notation for sensors is contradictory with respect to Lamb wave modes, please change the notation used for sensors to avoid it. Moreover, the fundamental explanation of Lamb wave mode is necessary.

6.     Please address to the computational duration using the presented approach and its applicability in real-world problems. This would be useful to add appropriate comments in these issues in conclusions as well.

7.     English need to be improved, numerous grammar and style errors are present in the manuscript.

Round 2

Reviewer 3 Report

The authors made appropriate clarifications in their answer to the review report and introduced necessary changes in the manuscript. From the scientific point of view I have no any additional comments and recommend the manuscript for a publication. The language editing is still necessary, including the new parts added after revision.